# The unhappy chaperone

Sara Linse[1]* 📖, Kyrre Thalberg[2] 📖 and Tuomas P. J. Knowles[3,4] 📖

[1]Biochemistry and Structural Biology, Chemical Centre, Lund University, Lund, Sweden; [2]Emmace Consulting, Lund, Sweden; [3]Yusuf Hamied Department of Chemistry, University of Cambridge, Cambridge, UK and [4]Cavendish Laboratory, University of Cambridge, Cambridge, UK

## Perspective

**Key words:**
Amyloid peptide solubility; chaperones; chemical potential; thermodynamics

**Author for correspondence:**
*Sara Linse,
E-mail: sara.linse@biochemistry.lu.se

### Abstract

Chaperones protect other proteins against misfolding and aggregation, a key requirement for maintaining biological function. Experimental observations of changes in solubility of amyloid proteins in the presence of certain chaperones are discussed here in terms of thermodynamic driving forces. We outline how chaperones can enhance amyloid solubility through the formation of heteromolecular aggregates (co-aggregates) based on the second law of thermodynamics and the flux towards equal chemical potential of each compound in all phases of the system. Higher effective solubility of an amyloid peptide in the presence of chaperone implies that the chemical potential of the peptide is higher in the aggregates formed under these conditions compared to peptide-only aggregates. This must be compensated by a larger reduction in chemical potential of the chaperone in the presence of peptide compared to chaperone alone. The driving force thus relies on the chaperone being very unhappy on its own (high chemical potential), thus gaining more free energy than the amyloid peptide loses upon forming the co-aggregate. The formation of heteromolecular aggregates also involves the kinetic suppression of the formation of homomolecular aggregates. The unhappiness of the chaperone can explain the ability of chaperones to favour an increased population of monomeric client protein even in the absence of external energy input, and with broad client specificity. This perspective opens for a new direction of chaperone research and outlines a set of outstanding questions that aim to provide additional cues for therapeutic development in this area.

## Towards a thermodynamic analysis of chaperone action

A large fraction of the proteins produced in the human body are chaperones that prevent other proteins from misfolding and aggregating (Bukau *et al.,* 2006; Hartl *et al.,* 2011, Balchin *et al.,* 2016). Such roles are key for maintaining biological function, and chaperones are critical components of organisms ranging from bacteria to human. Chaperones have been found to counteract protein aggregation and to solubilise protein aggregates already formed (Parsell *et al.,* 1994; Glover and Lindquist, 1998; Hageman *et al.,* 2010), for recent reviews see Zarrouchioti *et al.* (2017), Mogk *et al.* (2018), Nillegoda *et al.* (2018), Kampinga *et al.* (2019), Rosenzweig *et al.* (2019), Webster *et al.* (2019), Sinnige *et al.* (2020) and Chaplot *et al.,* (2020). Examples of chaperones that have been reported to inhibit different microscopic steps of amyloid formation *in vivo* or *in vitro* are HSP70 for tau (Kundel *et al.,* 2018; Nachman *et al.,* 2020), HSP70 and crystallins for α-synuclein (Dedmon *et al.,* 2005; Luk, 2008; Gaspar *et al.,* 2020), HSP70, DNAJB6 and DNAJB8 for poly-Q peptides (Muchowski *et al.,* 2000; Hageman *et al.,* 2010; Gillis *et al.,* 2013; Månsson *et al.,* 2014*b*; Kakkar *et al.,* 2016; Thiruvalluvan *et al.,* 2020), DNAJB proteins, Brichos and clusterin for amyloid β peptide (Yerbury *et al.,* 2007; Månsson *et al.,* 2014*a*, 2018; Cohen *et al.,* 2015) and HSP70, DNAJ and BiP for IAPP (Chien *et al.,* 2010). Anti-aggregation and disaggregase activities have also been observed for nanomaterials (Kim and Lee, 2003; Ikeda *et al.,* 2007; Cabaleiro-Lago *et al.,* 2008; Siposova *et al.,* 2020), polymers (Klajnert *et al.,* 2006; Song *et al.,* 2017), non-chaperone proteins (Assarsson *et al.,* 2014), micelle-forming surfactants (Cao *et al.,* 2007; Han *et al.,* 2010; He *et al.,* 2011) and small molecules (Habchi *et al.,* 2017; Li *et al.,* 2019).

Considerable effort is being devoted to understanding the effects of chaperones and co-chaperones on the mechanisms and rates of formation or disaggregation of amyloid fibrils (Cohen *et al.,* 2015; Arosio *et al.,* 2016; Karamanos *et al.,* 2019; Wentink *et al.,* 2020). From a molecular point of view, this information is contained in the heights of energy barriers separating monomeric and aggregated states, and the possible ways by which chaperones can influence such barriers. The effect of some chaperones on the end state of the aggregation process is less discussed and the current perspective regards this type of chaperone activity. In an equilibrium situation, the end state is independent of the path and related to the relative stabilities of the monomeric and aggregated states of proteins and chaperones. We present a thermodynamic analysis of different scenarios in systems composed of amyloid peptide and chaperone and discuss the experimental observations of changes in apparent solubility of amyloid proteins modulated by

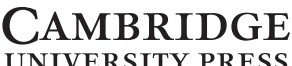

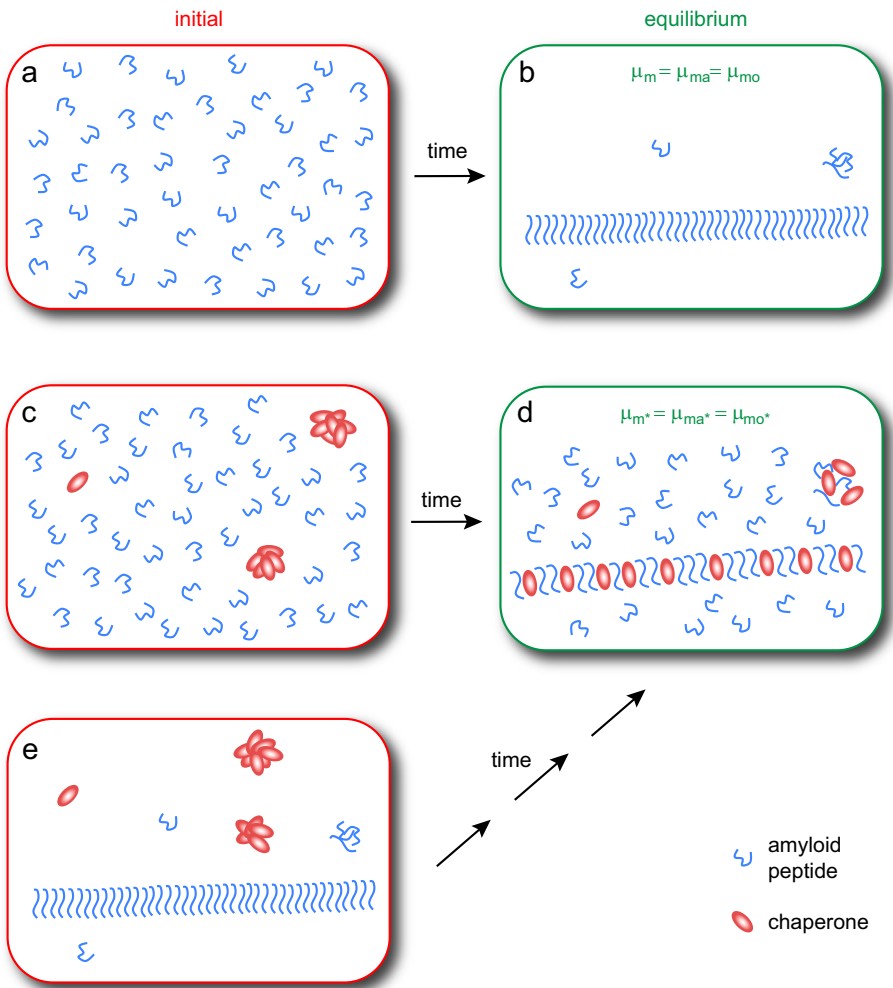

**Fig. 1.** Amyloid formation in the absence and presence of chaperone. In an initially supersaturated solution of amyloid peptide monomers (*a*), fibrillar aggregates will form until the chemical potential of monomers in solution is equal to the chemical potential of monomers in fibrillar and oligomeric aggregates, that is, until $\mu_m = \mu_{ma} = \mu_{mo}$ (*b*). In an initially supersaturated solution of amyloid peptide monomers and chaperone (*c*), aggregates will form until the chemical potential of monomers in solution is equal to the chemical potential of monomers in aggregates, that is, until $\mu_{m^*} = \mu_{ma^*} = \mu_{mo^*}$ (*d*), where * refers to the system of chaperone and peptide. At the end of the reaction, the chemical potential of chaperone monomers in solution is equal to the chemical potential of chaperone monomers in aggregates. If the system in panel *b* is supplemented with chaperone it is brought out of equilibrium (*e*) but has the same composition as the system shown in panel *c*. Thus, the same equilibrium state (*d*) will apply, although the time it takes to get there may be very long in absence of external energy input.

chaperones in terms of thermodynamic driving forces. We note that our discussion focuses on biologically relevant time scales and a living organism is never in a true equilibrium state as it is inherently an open system in which components are added and removed over time. Moreover, the interconversion between different polymorphs of aggregates, including between homomolecular and heteromolecular is often very slow, in analogy to solubility changes for different crystal polymorphs found for many small molecules. Many molecular processes are driven through the supply of external energy, and in particular many chaperone systems are ATP-dependent. However, the direction of change of a system is always towards lower free energy and we argue here that passive ATP-independent mechanisms are able to alter the effective protein solubility and discuss such mechanisms from a thermodynamic perspective, assuming in the following a constant temperature and pressure.

## Amyloid formation

As a consequence of the second law of thermodynamics, the chemical potential of a species, for example, an amyloid-forming peptide, at equilibrium is the same in every phase of the system regardless of how many phases are present (Evans and Wennerström, 1999; Atkins and DePaula, 2002). In a supersaturated solution of amyloid peptide monomers (Fig. 1*a*), aggregates will begin to form through nucleated polymerisation (Jarrett and Lansbury 1993; Knowles *et al.* 2009; Cohen *et al.*, 2011*a*, 2011*b*; Cohen *et al.* 2013) and the amount of fibrils will increase over time until the chemical potential of the monomers in solution, $\mu_m$, is equal to the chemical potential of monomers in fibrillar aggregates, $\mu_{ma}$ (Fig. 1*b*):

$$\mu_m = \mu_{ma}. \tag{1}$$

For the peptide monomer in solution, the chemical potential will depend both on its intrinsic free energy, $\varepsilon_m$, and the activity $a_m$:

$$\mu_m = \varepsilon_m + RT \ln a_m. \tag{2}$$

In Eq. (2), for pure systems the standard chemical potential $\mu_m^0$ is often used instead of $\varepsilon_m$.

In dilute solution, the activity can be approximated with the concentration [*m*]:

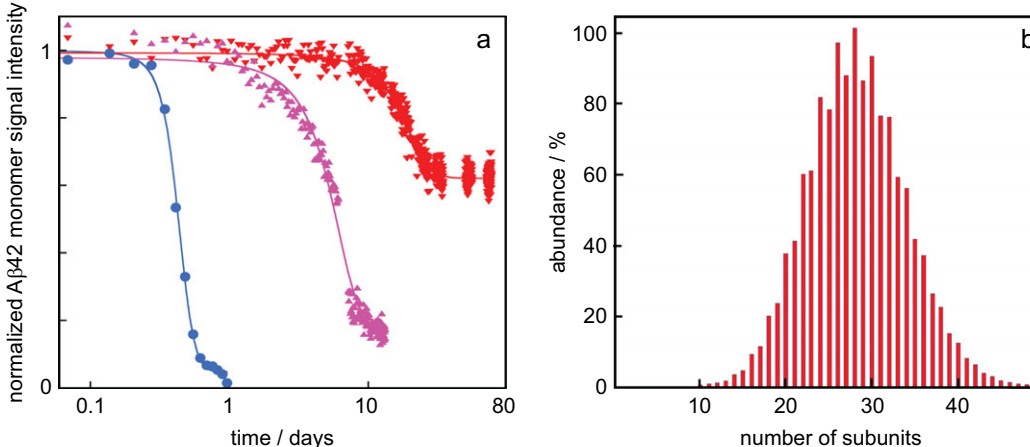

**Fig. 2.** Data from the literature. (*a*) Normalised Aβ42 monomer signal as a function of time as measured using solution nuclear magnetic resonance (NMR) spectroscopy for samples composed at time zero of 20 μM Aβ42 (blue circles), 20 μM Aβ42 + 2 μM DNAJB6 (red downward triangles) or 20 μM Aβ42 + 2 μM DNAJB6 mutant (purple upward triangles) in which 18 serine and threonine residues have been changed to alanines. These data are from (Månsson *et al.*, 2018) plotted with logarithmic *x*-axis. (*b*) Oligomer size distribution of the HSP70 chaperone as measured using mass spectrometry, replotted from data in Baldwin *et al.* (2011*a*).

$$\mu_m = \varepsilon_m + RT\,ln\,[m]. \tag{3}$$

The chemical potential of monomers in fibrils can be written as:

$$\mu_{ma} = \varepsilon_{ma} + RT\,ln\,[ma], \tag{4}$$

with $[ma]$ representing the concentration of monomers in fibrils and $\varepsilon_{ma}$ the intrinsic free energy of monomers in fibrils. Amyloid fibrils can be viewed as a discontinuous solid phase, but each aggregate is large, typically containing several thousands of monomers in identical highly ordered structure. Therefore, we can neglect the concentration-dependent translational entropy term and use the approximation:

$$\mu_{ma} = \varepsilon_{ma}. \tag{5}$$

At equilibrium (Fig. 1*b*), where monomers, oligomers and fibrils co-exist, the monomer concentration is equal to the solubility as follows by combining Eqs. (1), (3) and (5):

$$[m] = \exp\left\{\frac{\varepsilon_{ma} - \varepsilon_m}{RT}\right\}. \tag{6}$$

Even though the exchange rates may be low, the end state is a dynamic equilibrium in which all forward and backward reactions are still ongoing but at equal rates leading to no further change in the overall distribution of species.

Some of the monomers will be present in oligomers, that is, smaller aggregates of a different structure and growth rate than fibrillar aggregates (Dear *et al.*, 2020). At equilibrium, there will be a very low concentration of monomers in oligomers (Michaels *et al.*, 2020), but these will have the same chemical potential as monomers in fibrils and free monomer. Therefore, the discussion below regarding increased solubility needs to consider only the monomers and the larger fibrillar aggregates in the system.

### Chaperone induced suppression of amyloid formation and increase in peptide solubility

We now discuss how the thermodynamic framework can shed light on the origins of characteristic observations of amyloid peptide aggregation in the presence of certain chaperones, including, for

example, DNAJB1 and DNAJB6. Specifically, it is commonly observed that these chaperones (i) delay aggregation and (ii) elevate the peptide monomer concentration at the end of the reaction, compared to the situation in the absence of chaperone (Månsson *et al.*, 2014*a*, 2018; Schirmer *et al.*, 2016; Fig. 1*c,d*). For example, the aggregation of the amyloid β peptide from Alzheimer's disease, Aβ42, is significantly retarded in the presence of the chaperone DNAJB6 (Månsson *et al.*, 2014*a*, 2018) and the Aβ42 monomer concentration remains at much higher concentration after the reaction in the presence of DNAJB6 compared to a system with Aβ42 alone (Månsson *et al.*, 2018; Fig. 2*a*) in the absence of any external energy input. Increased solubility in the presence of chaperone has been reported in several other cases, for example, for tau fragments (Schirmer *et al.*, 2016), huntingtin (Scior *et al.*, 2018) and α-synuclein (Gao *et al.*, 2015) in the presence of HSP90, a tri-chaperone system and DNAJB1, respectively.

Crucially, experimental studies have failed to detect any direct interaction between DNAJB6 and Aβ42 monomers (Månsson *et al.*, 2014*a*, 2018), and an absence of strong interactions between the chaperone and amyloid peptide monomers appears to be a recurring feature (Dedmon *et al.*, 2005; Chien *et al.*, 2010; Österlund *et al.*, 2020). DNAJB6 inhibits nucleation of Aβ42 fibril formation, which has been interpreted in terms of interactions between the chaperone and oligomeric forms of the Aβ42 peptide (Månsson *et al.*, 2014*a*; Österlund *et al.*, 2020). Such a mechanism can explain the increased energy barrier for nucleation but not the shift of the final equilibrium state; as such, the elevated concentrations of peptide monomer at the end of the reaction in the presence of chaperone remains to be explained.

### The formation of co-aggregates can explain the increased amyloid peptide solubility in the presence of a chaperone

In dilute solutions in the presence of a chaperone, the intrinsic free energy, $\varepsilon_m$, of the free peptide monomer in solution remains unaltered, because it does not interact with the chaperone. The chemical potential of the monomer in solution in the presence of chaperone can therefore be written as:

$$\mu_{m*} = \varepsilon_m + RT\,ln\,[m_*], \tag{7}$$

where $*$ refers to the system of peptide and chaperone. The second law of thermodynamics must still hold, and when equilibrium is reached in the mixed system, monomers in solution and monomers in aggregates must have the same chemical potential (Fig. 1*d*):

$$\mu_{m*} = \mu_{ma*} = \varepsilon_{ma*}. \tag{8}$$

The second equality uses the same approximation as above Eq. (5). The solubility of the peptide in the presence of chaperone can then be expressed as:

$$[m*] = \exp\left\{\frac{\varepsilon_{ma*} - \varepsilon_m}{RT}\right\}, \tag{9}$$

and the observation that

$$[m_*] > [m], \tag{10}$$

implies that

$$\varepsilon_{ma*} > \varepsilon_{ma}. \tag{11}$$

Thus, the chaperone can only maintain an elevated concentration of peptide monomers at equilibrium if the formed aggregates are of a different structure in which the peptide has a higher chemical potential compared to the fibrils formed from peptide alone. If the solubility in the presence and absence of chaperone is known, it is possible to estimate the *difference* in chemical potential of the amyloid peptide in aggregates in the mixed compared to pure system:

$$\mu_{ma*} - \mu_{ma} = RT \ln\left(\frac{[m_*]}{[m]}\right). \tag{12}$$

An example of an aggregate of different structure would be a co-aggregate of peptide and chaperone (Fig. 1*d*). Several experimental observations can indeed be interpreted in terms of the formation of co-aggregates between amyloid peptides and chaperones (Dedmon *et al.*, 2005; Månsson *et al.*, 2014*a*; Gao *et al.*, 2015; Schirmer *et al.*, 2016; Zhang *et al.*, 2019). For example, the large aggregates formed in mixtures of Aβ42 and DNAJB6 were found to react with antibodies towards Aβ42 as well as antibodies towards DNAJB6 (Månsson *et al.*, 2014*a*). Other examples include co-aggregates of DNAJB1 with α-synuclein fibrils (Gao *et al.*, 2015) and of Hsp90 with tau fragments (Schirmer *et al.*, 2016). Chaperones are repeatedly observed in amyloid deposits *in vivo*, as reviewed by Sinnige and Morimoto (2020). While the formation of co-aggregates rather than pure fibrils would explain the increased solubility observed in the presence of chaperone, it also implies that the observed retardation of aggregate formation (Månsson *et al.*, 2018) involves the suppression of the appearance of a homomolecular aggregate phase on the time scale of the experiment.

### The unhappy chaperone

The increased effective solubility of the amyloid peptide in the presence of chaperone thus relies on the formation of aggregates in which the amyloid peptide has a higher chemical potential than in its pure aggregates. This is thermodynamically unfavourable for the amyloid peptide and we may ask why does it not form the more stable pure aggregates? To answer this question, we need to consider that it is the entire system of amyloid peptide and chaperone, which strives to minimise its free energy. This is schematically illustrated in Fig. 3.

The chemical potential of the chaperone can be expressed in terms of its intrinsic free energy, $\varepsilon_c$, and the chaperone concentration, $[c]$:

$$\mu_c = \varepsilon_c + RT \ln[c]. \tag{13}$$

The chemical potential of the chaperone in solution (although at low concentration) in the system of chaperone and peptide can be written as:

$$\mu_{c*} = \varepsilon_c + RT \ln[c_*], \tag{14}$$

and in analogy with the peptide, the chemical potential of the chaperone in the aggregates formed in the system with peptide and chaperone is:

$$\mu_{ca*} = \varepsilon_{ca*}. \tag{15}$$

Crucially, an overall reduction in free energy is possible only if the chaperone has a higher chemical potential alone than in the co-aggregates that form in the system of peptide and chaperone (Fig. 3*b*). The chaperone then 'gains' more by forming the co-aggregate than the amyloid peptide 'loses', that is, the chaperone lowers its chemical potential more than the amyloid peptide increases its chemical potential (cf. states iii and iv in Fig. 3*b*). We can thus predict that the more unhappy the chaperone is on its own, the more prone will it be to interact with the amyloid peptide and the more potent will it be in increasing the solubility of the amyloid peptide. The net reduction in free energy will rely on the relative stoichiometry of chaperone to amyloid peptide in the co-aggregates and their total concentrations in the system as a whole. Thus, the higher the fraction of chaperone in the co-aggregate, the higher the increase in peptide solubility. A challenge for the future will be to establish the limits, if any, of chaperone to peptide stoichiometries in co-aggregates.

### Analogous systems: co-crystallisation and co-micelle formation

Within pharmaceutical science, the poor solubility of a drug molecule can be remediated through co-crystallisation with another substance (Gagniere, 2009; Schartman *et al.*, 2009; Zhang and Rasmussen, 2013). The co-crystal is of different structure compared to the pure drug crystal and the drug monomer in the co-aggregate has higher chemical potential than the drug in the pure crystal. This leads to higher drug solubility in the mixed, $[m*]$, compared to pure system, $[m]$. In a binary surfactant mixture, one surfactant typically gains from mixed micelle formation, while the other one loses, at constant total concentration (Nagarajan, 1985). Again, it is the system as a whole that strives to minimise its free energy and the increase in chemical potential of one component is compensated for by a larger reduction for the other component.

### The role of crystal and fibril morphology

Eq. (6) implies that the more stable the aggregate structure, the lower the apparent solubility. For a give compound, there may be more than one crystalline structure that can form, with different apparent solubility (Brittain, 2009). In analogy, some amyloid peptides can end up in more than one morphology (Lutter *et al.*, 2019; Ke *et al.*, 2020). In such cases, the different crystal forms or morphologies are separated by kinetic barriers. Solubility measurements may reveal which of the aggregate forms is closest to the

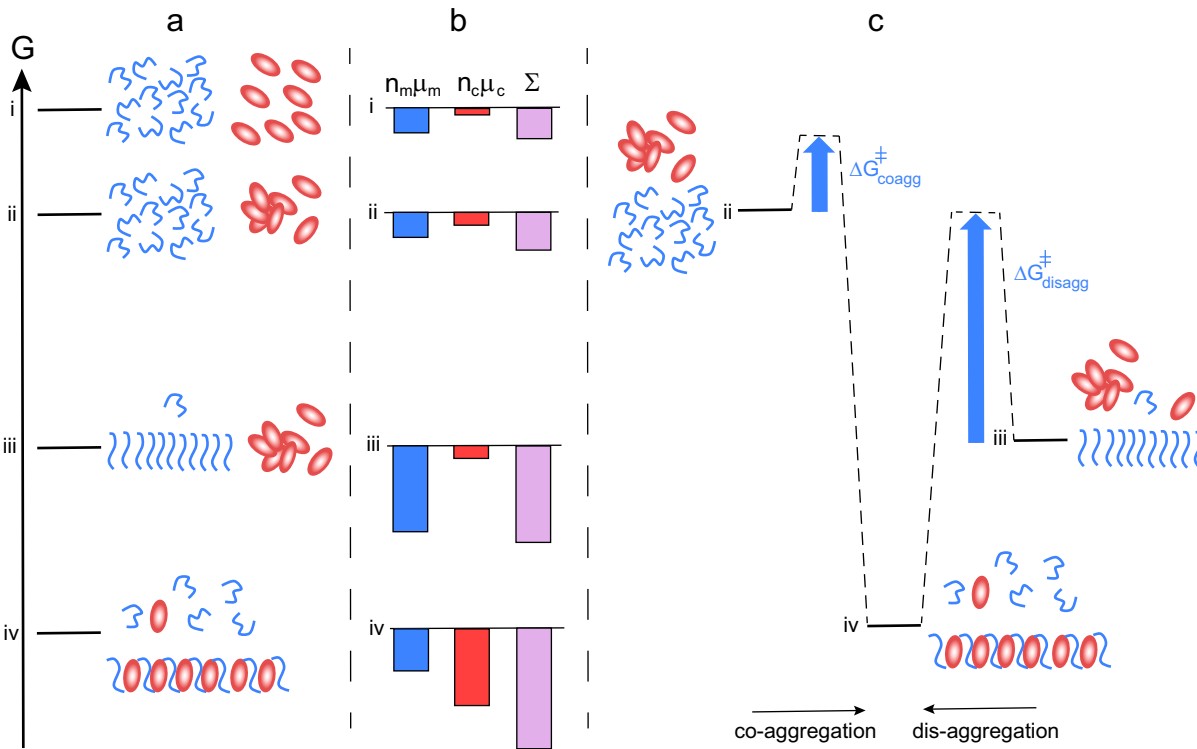

**Fig. 3.** Thermodynamics of co-aggregation and disaggregation. (*a*) Free energy diagram for a closed system of an amyloid peptide (blue) and chaperone (red). At the highest level i, all species are monomeric. At level ii, the peptide is monomeric and the chaperone a mixture of monomers and oligomers. At level iii, there are amyloid fibrils, chaperone oligomers and monomers of both, and the peptide monomer concentration in this state is higher than in state iii. (*b*) Chemical potential summed up over all molecules of each kind separately (peptide blue and chaperone red), and for the system (purple). There is a large increase in amyloid peptide solubility from state iii to state iv. (*c*) Reaction coordinate diagram for the case of co-aggregation (from ii to vi) and for disaggregation (from iii to iv) illustrating the much higher energy barrier in the latter case.

thermodynamic minimum, because the more stable the crystal or fibril, the lower the solubility.

### The role of chaperone oligomerisation

Common characteristics for ATP-independent chaperones are their tendency to form: (i) polydisperse (Baldwin *et al.,* 2011*a*; Månsson *et al.,* 2014*b*; Fig. 2*b*) and (ii) relatively disordered aggregates, in which the buried surface area at contact points is small (Baldwin *et al.,* 2011*b*; Söderberg *et al.,* 2018), and with rapid exchange between monomers and aggregates of various sizes (Luk, 2008; Månsson *et al.,* 2014*b*; Karamanos *et al.*; 2019). These clusters are typically stable relative to the monomeric form of the chaperone, and the free monomer concentration is too low to be detected by most techniques. One may argue that both features are requirements for the chaperone action.

The observed very low chaperone monomer concentration is necessary for its action because a protein with low chemical potential on its own would be rather happy and the reduction in chemical potential upon forming a co-aggregate would not be enough to compensate for the amyloid peptide's increase in chemical potential. Moreover, the presence of chaperone polydisperse clusters means that there is no well-defined aggregation number, contrary to the case of surfactant micelles. This means that no arrangement can satisfy the chaperone, thus its chemical potential in the aggregates is still high. In contrast, non-chaperone proteins like for example haemoglobin and virus capsids form aggregates with a well-defined aggregation number, which are relatively stable. Polydispersity thus

seems to be a key attribute enabling the broad range of peptide substrates that chaperones can interact with and remedy. It seems straight-forward to predict that the chemical potentials of chaperones are higher than for most proteins, that is, chaperones are more unhappy. A challenge for the future will be to establish quantitative estimates of this difference.

### Less potent chaperones variants

It follows from above that the happier the chaperone is on its own, the less potent it will be in increasing the solubility of the amyloid peptide. In this context, we may discuss mutations and substitutions that alter the potency of a chaperone. For example, in the case of DNAJB6, substitutions from Ser or Thr to Ala, or removal of major parts of the Ser/Thr-rich region, lead to chaperones that are less potent in inhibiting Aβ42 aggregation and the solubility of Aβ42 is closer to that of Aβ42 alone (Månsson *et al.,* 2018; Fig. 2*a*). Such results are often discussed in terms of altered interaction between chaperone and client peptide, and we argue here that it may be fruitful to consider also the effect on the chaperone chemical potential (happiness). An increased solubility of the amyloid peptide would result if the substitutions make the chaperone less unhappy, that is, reduce the difference in chemical potential of chaperone on its own relative to in the co-aggregate. In support of this view, removal of the Ser/Thr-rich region was found to increase the proportion of monomeric DNAJB6 (Karamanos *et al.,* 2019) suggesting that the variant is more happy on its own compared to the very unhappy wild-type chaperone.

### Fibril disaggregation upon change in solution conditions

Self-assembly is a dynamic process. The simplest way to disaggregate fibrils *in vitro* is therefore by dilution (Nespovitaya, 2016; Schirmer *et al.,* 2016). This brings the monomer concentration below the solubility limit, and the net effect will be a dissolution of fibrils and an increase in monomer concentration until equilibrium is re-established (Eq. (1)). Other means of disaggregating fibrils are by adjustment of the temperature or pH to regimes where the solubility is higher. Fibril disassembly may also be facilitated by the addition of cationic surfactants (He *et al.*, 2011), polymers (Song *et al.*, 2017), fullerenes (Siposova *et al.*, 2020) or small molecules (Li *et al.*, 2019), some of which form co-aggregates of altered structure compared to the peptide-only fibrils.

### The role of chaperones in passive disaggregation

*In vivo*, disaggregation is facilitated by a class of chaperones, or chaperone systems, called disaggregases (Glover and Lindquist, 1998; Gao *et al.*, 2015; Kityk *et al.,* 2018; Mogk *et al.,* 2018; Nachman *et al.,* 2020). If a disaggregase is added to a sample containing amyloid fibrils, the fibril concentration is observed to decrease (see, e.g. Gao *et al.*, 2015; Nachman *et al.*, 2020; Wentink *et al.*, 2020). Disassembly is typically incomplete because it cannot proceed beyond the point where equilibrium is re-established (Eq. (1)). While most disaggregases studied to date are ATP-dependent, the framework in this paper allows us to consider a purely passive mode of action. Adding a chaperone to a solution of amyloid peptide fibrils, in dynamic equilibrium with monomers and oligomers puts the system out of equilibrium as shown in Fig. 1*e*. The net change, whereby the system establishes a new equilibrium, will be a dissolution of peptide-only fibrils and formation of co-aggregates until the chemical potential of monomers in solution is equal to the chemical potential of monomers in the co-aggregates, that is, $\mu_{m^*} = \mu_{ma^*}$. In Fig. 3, this corresponds to a change from state iii to iv. To satisfy the condition of reversibility, the system should reach the same end state regardless of the initial species distribution; the systems in Fig. 1*c,e* have the same total composition and will therefore have the same equilibrium, depicted in Fig. 1*d.*.

Passive disaggregation in the presence of chaperones will thus occur if the intrinsic free energy (equilibrium structure) of aggregates formed in the presence of the chaperone is different from those formed in its absence. This allows for a shift in equilibrium and a net dissolution of fibrils. However, the energy barrier separating the system from the equilibrium state can be very high (Fig. 3*c*, right *vs* left) meaning that equilibrium may not be reached over any reasonable experimental time frame.

### The role of chaperones in active disaggregation

Nature has solved this by utilising ATP hydrolysis; the disaggregase activity in multi-cellular organisms is commonly provided by tri-chaperones systems composed of HSP70, HSP40 (J-domain protein) and a nucleotide-exchange factor (Kampinga and Craig, 2010; Shorter, 2011; Rampelt *et al.*, 2012; De Los Rios and Barducci, 2014; Nillegoda *et al.*, 2015; Kityk *et al.*, 2018; Mogk *et al.*, 2018; Rosenzweig *et al.*, 2019; Faust *et al.*, 2020; Nachman *et al.*, 2020). The disaggregase activity of HSP70 is coupled to ATP hydrolysis, meaning that the free energy of dephosphorylation of ATP is 'invested' to speed up or drive the disaggregation further than would otherwise be possible (De Los Rios and Barducci,

2014). The nucleotide-exchange factor secures continuous activity through the restoration of the ATP-binding form of HSP70.

Many studies have identified the strict requirement for a J-domain protein, for disaggregation to happen, although the energy input relies on HSP70 and nucleotide-exchange factor to lower the kinetic barrier of disassembly. While the role of the J-domain protein is concluded to provide specificity (Kampinga *et al.*, 2019), we argue that an additional and equally important role of the J-domain may be to form the co-aggregate with the amyloid peptide. The ability of a functional disaggregase machinery to change the solubility (equilibrium) requires that at least one component forms a co-aggregate with the amyloid peptide, organised in a manner such that the peptide has a higher chemical potential compared to the starting fibril. If no co-aggregates were formed, the peptide monomers that are released would re-elongate the existing fibrils or form new fibrils of the same structure and chemical potential as before and no increase in solubility would be possible.

### Fibril coating

The mechanism outlined in this perspective relies on the formation of co-aggregates of chaperone and amyloid peptides, which are significantly different from those of the peptide alone. This mode of action is thus very different compared to some other classes of chaperones that merely coat fibrils while leaving the fibril structure largely intact (Cohen *et al.*, 2015; Fig. 4). In such cases, the equilibrium position would be shifted towards lower solubility of the monomer compared to the situation in the absence of chaperone (Fig. 4, level iv *vs* iii). Fibril coating can, however, explain the reduced rate of fibril-catalysed secondary nucleation in the presence of this class of chaperones or substances (Cohen *et al.*, 2015; Arosio *et al.*, 2016; Habchi *et al.*, 2017; Linse *et al.*, 2020). In both the scenario where the chaperone and amyloid peptide forms co-aggregates of altered structure (Fig. 3) and the scenario where fibrils are simply coated with the chaperone (Fig. 4), the system as a whole reduces its free energy relative to the mix of chaperone oligomers and amyloid fibrils, state iii. However, only the formation of co-aggregates of altered structure can result in increased solubility of the amyloid peptide.

### Concluding remarks and perspective

Many proteins, chaperones and others alike, retard amyloid formation through interactions with discrete species along the reaction pathway, which increase the energy barriers for nucleation or elongation of aggregates. Such processes are driven through the free energy gain of the chaperone binding to the target aggregate structure and can take place even in the absence of external energy input. However, a change in the solubility (equilibrium) of the amyloid peptide can only be achieved, in the absence of external energy input, by a particular class of chaperones that are very unhappy on their own. This class includes for example several of the DNAJB proteins and αB crystallins. The action of these chaperones in the suppression of aggregation and promotion of disaggregation may be understood in terms of the second law of thermodynamics and the equal chemical potential in all phases of a compound at thermodynamic equilibrium. An increased solubility of the amyloid peptide can thus only be at hand if the peptide is engaged in an alternative form of aggregates in which its chemical potential is higher than in the regular fibrils. The formation of heteromolecular aggregates, that is, co-aggregates of chaperone and

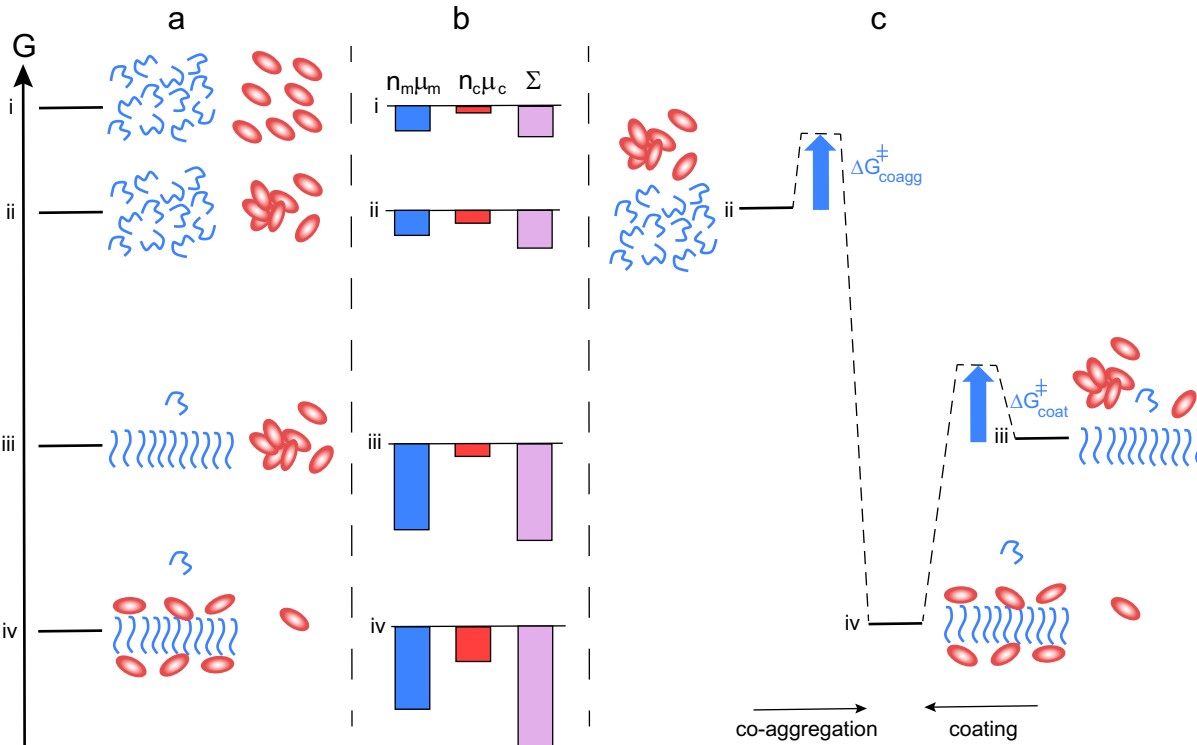

**Fig. 4.** Thermodynamics of aggregation and fibril coating. (*a*) Free energy diagram for a closed system of an amyloid peptide (blue) and chaperone (red). At the highest level i, all species are monomeric. At level ii, the peptide is monomeric and the chaperone a mixture of monomers and oligomers. At level iii, there are amyloid fibrils, chaperone oligomers and monomers of both. At level iv, peptide fibrils coated by chaperone co-exist with monomers of both, and the peptide monomer concentration in this state is similar or lower than in state iii. (*b*) Chemical potential summed up over all molecules of each kind separately (peptide blue and chaperone red), and for the system (purple). In this case, there is no increase in solubility from state iii to state iv. (*c*) Reaction coordinate diagram for the case of co-aggregation leading to coated fibrils (from ii to vi) and coating of already existing fibrils (from iii to iv).

amyloid peptide, may lead to enhanced peptide solubility, in analogy with the increase in small molecule solubility upon formation of co-crystals. The key driver is the high (unfavourable) chemical potential of the chaperone on its own, which is lowered upon formation of a co-aggregate. The gain for the chaperone needs to exceed the increase in chemical potential (loss) of the amyloid peptide such that the formation of co-aggregates lowers the free energy of the system as a whole.

Unhappiness and polydispersity are key attributes of this class of chaperones enabling the broad range of peptide substrates that these chaperones can interact with and remedy. Consideration of the thermodynamics of chaperone actions as well as energy barriers can lead to improved understanding of the passive role of chaperones as aggregation inhibitors as well as active disaggregation fueled by an external energy input. We outline here a number of outstanding questions that may be addressed in future experiments to provide qualitative insights and quantitative measures of key parameters regarding the action of unhappy chaperones. Such research may forward our understanding of this class of chaperones and may provide clues towards the prevention of neurodegenerative and other amyloid diseases.

- What is the composition (molar ratio of chaperone to amyloid peptide) in the final co-aggregates as a function of total concentrations of chaperone and peptide? Are there upper and lower limits of the possible stoichiometric ratios of chaperone and peptide? This may be addressed using mass spectrometry with isotope standards.

- What is the structure of the final co-aggregates on ultra-structural level and in atomic detail? Are the co-aggregates interdigitated fibrils with regularly alternating peptide and chaperone, or do they have some other structural arrangement? This may be addressed using cryo-electron microscopy; atomic force microscopy, nano-IR and solid-state nuclear magnetic resonance (NMR) spectroscopy as well as neutron scattering with contrast variation.

- How does the self-aggregation and polydispersity of the chaperone alone vary with total concentration? How does the self-aggregation and polydispersity of less potent chaperone variants vary with total concentration? This may be addressed using diffusion measurements in microfluidics, fluorescence correlation spectroscopy, analytical ultracentrifugation or quantification of monomer after separation from aggregates.

- Can the chaperone gain in free energy upon co-aggregation be quantified? This can be addressed using fractionation methods and spectroscopic or isotope-based methods to provide quantitative numbers on species distributions. The amount of unbound chaperone may be used to estimate the free energy gain for the chaperone due to co-aggregation.

- Which molecular features make the chaperone unhappy? Chaperones are typically folded into discrete domains, implying that there is a very strong driving force for the formation of these domains, as they are formed despite of exposing energetically unfavourable exterior surfaces. Are the exteriors evolved in a way that they will not match with any other chaperone domain surfaces, to avoid high-affinity self-association?

- How much does water contribute to the unusually high chemical potential of chaperones? Chaperones seem unable to bury most of their hydrophobic patches by oligomerisation. This may be addressed in a quantitative manner using temperature-dependent measurements to estimate the heat capacity change of chaperone association.
- How much do the disordered termini and linkers contribute to the unusually high chemical potential of chaperones?
- What is the time scale for the system to relax towards an equilibrium distribution and structures in two systems with the same total composition, but with different starting points, for example, chaperone plus peptide monomer *versus* chaperone plus peptide fibrils? Will in the latter case equilibrium be achieved during any reachable time frame in the absence of external energy input? This may be addressed using NMR spectroscopy and ThT fluorescence to reveal the relative height of the barriers (activation energies) for aggregation and disaggregation.

**Acknowledgements.** We are grateful to Daan Frenkel for stimulating discussions and input on thermodynamics of multiphase systems. This work was supported by the Swedish Research Council VR (2015-00143 to S.L.) and through the European Research Council under the European Union's Seventh Framework Programme (FP7/2007-2013) through the ERC grant (to T.K.) PhysProt (agreement no. 337969).

**Open Peer Review.** To view the open peer review materials for this article, please visit http://dx.doi.org/10.1017/qrd.2021.5.

**Conflict of interest.** The authors declare no conflicts of interest.

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
