## [Reviewer Report]

*Comments to Author*: The authors build up on an interesting observation that is prevalent in literature i.e. the equilibrium level of aggregated protein in amyloid state is different in the presence of chaperones. I believe that they try to base their formulation towards explaining this observation. While most of the literature has focused on the kinetics, this is a novel approach. I have however several queries which bely my understanding of the subject and would require the authors to clarify this before I can look at the bigger picture.

1.The authors have used reversible thermodynamic equilibrium equations (equality of chemical potential) to argue in equation 8 that the chemical potential needs to be same. Am I mistaken in this assumption?

2.Is there any evidence that amyloid formation is reversible in vitro under conditions aggregation ismonitored? Amyloids are notoriously difficult to solubilize and hence I was under the impression that this is most likely - close to - an irreversible process. How would the formulation take care of this problem?

3.How would equation 11 true in a co-aggregate? It is not a pure state. Equation 7 is also based on the assumption that the chemical potential in the co-aggregate is the standard chemical potential! They are not pure states and interactions will change the chemical potential of both the species significantly. This may be one of the guiding factors. Is there something that I am missing out here?

Isnt it more likely that the chaperones prevent or bind to a conformation that is monomeric and capable of aggregation? It has been shown using multiple kinetics data that there is a unimolecular nucleation phase that may indicate the formation of conformations that are aggregation prone. How is this considered in the manuscript? How will the change in structures of the monomer be captured by chemical potential equations?

Although the authors literature evidence that the monomers do not bind to the chaperones, there also has been many evidence to show that that it binds to non-native conformations. Conformations that are enroute to formation of aggregation nucleus. This binding may shift the equilibrium back towards the folded state, although I am still intrigued (as much as the authors are) with published evidence that the chaperones can change the thermodynamics. Similar things happens during folding (see below).

The analysis is straightforward and is not scientifically very novel, although the approach of looking at this problem is.

If we look at chaperone (holdase) assisted protein folding, we see an analogous situation. Given that Chaperones bind to non-native conformations, and bind to them very stably, the free energy of interaction is negative, thereby rewording the interaction as the "chaperones being unhappy" is interesting - but does it tell us something. During folding (while being assisted by holdases), the native protein does not bind to the chaperones, while the non-native protein does, and in doing so the chaperone is able to shift the equlilbrium towards the native state, which it does not bind to. The same case seems to be true here. The situation there needs multiple steps to explain the process, including rate of spontaneous folding. Additionally, this is thought to happen as chaperones can prevent pathways that are non-productive, aggregation, or increase the rate of folding thereby changing the equilibrium constant of folding. Is it possible that a similar framework may describe even the whole aggregation process more succinctly than described in the manuscript?

Why should I not be able to argue this way: Assuming that the co-aggregate is less stable than the pure aggregate of the amyloid (since the equilibrium is shifted in the presence of chaperones), the monomers of aggregation prone protein may be at the same chemical potential in the pure and mixed aggregate while the chaperones face more cost in chemical potential while aggregating and hence destabilizes the aggregate by increasing the free energy of the co-aggregate over the pure aggregate. The authors believe that chaperones are actually stabilizing the co-aggregate (coz they are unhappy alone), which should mean that the aggregates are actually more stable in the presence of chaperones! I believe the claim is that the co-aggregates are more favorable over pure aggregates, but under these conditions the aggregating monomers are less stable than when they aggregate alone. Thereby the monomers remain more in solution. But then, shouldn’t a small amount of aggregation prone monomer be catalytic in aggregating most of the chaperone protein?

---

## [Reviewer Report]

*Comments to Author*: This is a thoughtful paper offering an intuitive and rigorous explanation for the tendency of chaperones to increase the solubility of monomeric + oligomeric forms of amyloidogenic peptides relative to the fibril form.The concept of an unhappy chaperone is appealing and I hope will be widely used.The arguments are laid out in a logical way using equations familiar to anyone how has had first-year physical chemistry, without requiring an advanced concepts in thermodynamics.I have only a few comments:

If I understand correctly the definition of the concentration of the soluble form would be the sum of the activity of the monomeric peptide in all forms that are soluble.This should be made more clear, if I have this correctly, because association with chaperones can increase the pool of oligomeric forms.

In the analogy to drug crystal forms, it should be made explicit that there is generally not one crystalline form for any given compound, so the apparent solubility really depends on the crystalline form (may be under kinetic control).Also, it might be easier conceptually for readers to think of different salt forms having different crystal lattices and interactions, and hence different solubilities.

Like the analogy with salt forms, the apparent solubility of a monomer will depend on its equilibrium with the specific fibrillar form.

p. 8:par 1, "through the restoration of ATP from ADP" should make clear the conformational forms of the protein binding to ATP vs. ADP is what is meant.

eq. 10 seems better to have RT rather than kT for continuity with other equations?

---

## [Reviewer Report]

*Comments to Author*: The very well written manuscript entitled "the unhappy chaperone" by the Linse/Knowles "consortium" describes the protein aggregation interference of chaperones from a thermodynamics point of view, which is on the one hand straight forward, but on the other hand an important contribution in the elucidation of the disaggregation mechanism of chaperones.The reviewer has several points that are up for discussion:

(i)Often amyloid fibrils are considered as kinetically trapped. How is this notion influencing the presented theoretical work.

(ii)The thermodynamics presented is under constant temperature and pressure, which should be mentioned.

(iii)Why can the fibril considered a pure state? Is it because it is assumed to be a solid with activity 1?

(iv)Similarly, the chemical potential of the chaperone in the co-aggregates is considered equal to the standard chemical potential. Why is this the case?

(v)The reviewer invites the authors to elaborate some more on the important sentence that "The net reduction in free energy will rely on the relative stoichiometry of the chaperone and amyloid peptide in the co-aggregates and their total concentrations in the system as a whole"

(vi)The authors in the manuscript do not entirely split the microscopic description of nature with the macroscopic description put forward here. For example: "Such results are often discussed in terms of altered interaction between chaperone and client peptide, but we argue here that it may be fruitful to consider the effect on the chaperone chemical potential (happiness)."It is not about the one or the other. The macroscopic picture put forward here does not exclude the microscopic interpretation given.

(vii)The presented theory is concentrating on a liquid solid phase transition, but there might also be the possibility of a liquid liquid phase separation if the chaperone acts on the disaggregation of the amyloid.

(viii)It is indicated that the unhappy chaperone may also be unhappy because of the water and thus the role of water in chaperone activity is an interesting point to be studied (could be mentioned in the question list).

---

## [Reviewer Report]

*Comments to Author*: Dear Drs. Linse, Thalberg and Knowles,

Your manuscript entitled "The unhappy chaperone" has been reviewed by 3 referees. The reviewers were of the opinion that the manuscript contains important information of interest to other investigators. All three reviewers have very good suggestions how to further improve the manuscript, and questions that need to be resolved.

Please modify the manuscript to address all of the reviewers’ questions and submit the revised document via the QRB-D Electronic Submission site. I would appreciate if you also upload your source files and a separate letter that includes a point-by-point response to all of the issues raised in the reviews. Review of your revised manuscript will be facilitated by uploading a copy of the original manuscript marked with the changes using Track Changes (MS Word), highlighting, or colored text.

Reviews:

Reviewer #1: This is a thoughtful paper offering an intuitive and rigorous explanation for the tendency of chaperones to increase the solubility of monomeric + oligomeric forms of amyloidogenic peptides relative to the fibril form.The concept of an unhappy chaperone is appealing and I hope will be widely used.The arguments are laid out in a logical way using equations familiar to anyone how has had first-year physical chemistry, without requiring an advanced concepts in thermodynamics.I have only a few comments:

If I understand correctly the definition of the concentration of the soluble form would be the sum of the activity of the monomeric peptide in all forms that are soluble.This should be made more clear, if I have this correctly, because association with chaperones can increase the pool of oligomeric forms.

In the analogy to drug crystal forms, it should be made explicit that there is generally not one crystalline form for any given compound, so the apparent solubility really depends on the crystalline form (may be under kinetic control).Also, it might be easier conceptually for readers to think of different salt forms having different crystal lattices and interactions, and hence different solubilities.

Like the analogy with salt forms, the apparent solubility of a monomer will depend on its equilibrium with the specific fibrillar form.

p. 8:par 1, "through the restoration of ATP from ADP" should make clear the conformational forms of the protein binding to ATP vs. ADP is what is meant.

eq. 10 seems better to have RT rather than kT for continuity with other equations?

Reviewer #2: The very well written manuscript entitled "the unhappy chaperone" by the Linse/Knowles "consortium" describes the protein aggregation interference of chaperones from a thermodynamics point of view, which is on the one hand straight forward, but on the other hand an important contribution in the elucidation of the disaggregation mechanism of chaperones.The reviewer has several points that are up for discussion:

(i)Often amyloid fibrils are considered as kinetically trapped. How is this notion influencing the presented theoretical work.

(ii)The thermodynamics presented is under constant temperature and pressure, which should be mentioned.

(iii)Why can the fibril considered a pure state? Is it because it is assumed to be a solid with activity 1?

(iv)Similarly, the chemical potential of the chaperone in the co-aggregates is considered equal to the standard chemical potential. Why is this the case?

(v)The reviewer invites the authors to elaborate some more on the important sentence that "The net reduction in free energy will rely on the relative stoichiometry of the chaperone and amyloid peptide in the co-aggregates and their total concentrations in the system as a whole"

(vi)The authors in the manuscript do not entirely split the microscopic description of nature with the macroscopic description put forward here. For example: "Such results are often discussed in terms of altered interaction between chaperone and client peptide, but we argue here that it may be fruitful to consider the effect on the chaperone chemical potential (happiness)."It is not about the one or the other. The macroscopic picture put forward here does not exclude the microscopic interpretation given.

(vii)The presented theory is concentrating on a liquid solid phase transition, but there might also be the possibility of a liquid liquid phase separation if the chaperone acts on the disaggregation of the amyloid.

(viii)It is indicated that the unhappy chaperone may also be unhappy because of the water and thus the role of water in chaperone activity is an interesting point to be studied (could be mentioned in the question list).

Reviewer #3: The authors build up on an interesting observation that is prevalent in literature i.e. the equilibrium level of aggregated protein in amyloid state is different in the presence of chaperones. I believe that they try to base their formulation towards explaining this observation. While most of the literature has focused on the kinetics, this is a novel approach. I have however several queries which bely my understanding of the subject and would require the authors to clarify this before I can look at the bigger picture.

1.The authors have used reversible thermodynamic equilibrium equations (equality of chemical potential) to argue in equation 8 that the chemical potential needs to be same. Am I mistaken in this assumption?

2.Is there any evidence that amyloid formation is reversible in vitro under conditions aggregation ismonitored? Amyloids are notoriously difficult to solubilize and hence I was under the impression that this is most likely - close to - an irreversible process. How would the formulation take care of this problem?

3.How would equation 11 true in a co-aggregate? It is not a pure state. Equation 7 is also based on the assumption that the chemical potential in the co-aggregate is the standard chemical potential! They are not pure states and interactions will change the chemical potential of both the species significantly. This may be one of the guiding factors. Is there something that I am missing out here?

Isnt it more likely that the chaperones prevent or bind to a conformation that is monomeric and capable of aggregation? It has been shown using multiple kinetics data that there is a unimolecular nucleation phase that may indicate the formation of conformations that are aggregation prone. How is this considered in the manuscript? How will the change in structures of the monomer be captured by chemical potential equations?

Although the authors literature evidence that the monomers do not bind to the chaperones, there also has been many evidence to show that that it binds to non-native conformations. Conformations that are enroute to formation of aggregation nucleus. This binding may shift the equilibrium back towards the folded state, although I am still intrigued (as much as the authors are) with published evidence that the chaperones can change the thermodynamics. Similar things happens during folding (see below).

The analysis is straightforward and is not scientifically very novel, although the approach of looking at this problem is.

If we look at chaperone (holdase) assisted protein folding, we see an analogous situation. Given that Chaperones bind to non-native conformations, and bind to them very stably, the free energy of interaction is negative, thereby rewording the interaction as the "chaperones being unhappy" is interesting - but does it tell us something. During folding (while being assisted by holdases), the native protein does not bind to the chaperones, while the non-native protein does, and in doing so the chaperone is able to shift the equlilbrium towards the native state, which it does not bind to. The same case seems to be true here. The situation there needs multiple steps to explain the process, including rate of spontaneous folding. Additionally, this is thought to happen as chaperones can prevent pathways that are non-productive, aggregation, or increase the rate of folding thereby changing the equilibrium constant of folding. Is it possible that a similar framework may describe even the whole aggregation process more succinctly than described in the manuscript?

Why should I not be able to argue this way: Assuming that the co-aggregate is less stable than the pure aggregate of the amyloid (since the equilibrium is shifted in the presence of chaperones), the monomers of aggregation prone protein may be at the same chemical potential in the pure and mixed aggregate while the chaperones face more cost in chemical potential while aggregating and hence destabilizes the aggregate by increasing the free energy of the co-aggregate over the pure aggregate. The authors believe that chaperones are actually stabilizing the co-aggregate (coz they are unhappy alone), which should mean that the aggregates are actually more stable in the presence of chaperones! I believe the claim is that the co-aggregates are more favorable over pure aggregates, but under these conditions the aggregating monomers are less stable than when they aggregate alone. Thereby the monomers remain more in solution. But then, shouldn’t a small amount of aggregation prone monomer be catalytic in aggregating most of the chaperone protein?

---

## [Reviewer Report]

*Comments to Author*: Reviewer #1: The paper is acceptable for publication

Reviewer #2: The authors have addressed all the queries of the reviewer